# Third-Generation Dynamic Anterior Plate-Screw System for Quadrilateral Fractures: Digital Design Based on 834 Pelvic Measurements

**DOI:** 10.3390/medicina59020211

**Published:** 2023-01-21

**Authors:** Ranran Shang, Haiyang Wu, Li Zhou, Chengjing Song, Qipeng Shao, Ximing Liu, Xianhua Cai

**Affiliations:** 1Department of Orthopaedic Surgery, Wuhan Hospital of Integrated Traditional Chinese and Western Medicine, No.215 Zhongshan Avenue, Wuhan 430031, China; 2Department of Clinical Medicine, Graduate School of Tianjin Medical University, No.22 Qixiangtai Road, Tianjin 301700, China; 3Duke Molecular Physiology Institute, Duke University School of Medicine, Duke University, Durham, NC 27708, USA; 4Department of Rehabilitation Medicine, Wuhan Hospital of Integrated Traditional Chinese and Western Medicine, No.215 Zhongshan Avenue, Wuhan 430031, China; 5Department of Orthopaedic Surgery, The Fifth People’s Hospital of Huaian, Xiangjiang Road, Huaian 223002, China; 6Department of Joint Surgery, People’s Hospital of Ganzhou, No.16 Mei Guan Avenue, Ganzhou 341000, China; 7Department of Orthopaedic Surgery, General Hospital of Central Theater Command, No.627 Wuluo Road, Wu Chang District, Wuhan 430030, China; 8Department of Orthopaedic Surgery, Huanan Hospital, Shenzhen University, No.1 Fuxin Road, Shenzhen 518060, China

**Keywords:** acetabular fracture, quadrilateral, fracture fixation, digital measurement, instrumentation

## Abstract

*Background and Objectives*: To investigate the digital measurement method for the plate trajectory of dynamic anterior plate-screw system for quadrilateral plate (DAPSQ), and then design a third-generation DAPSQ plate that conforms to the needs of the Chinese population through collating a large sample anatomical data. *Materials and Methods*: Firstly, the length of the pubic region, quadrilateral region, iliac region, and the total length of the DAPSQ trajectory were measured by a digital measurement approach in 22 complete pelvic specimens. Then, the results were compared with the direct measurement of pelvic specimens to verify the reliability of the digital measurement method. Secondly, 504 cases (834 hemilateral pelvis) of adult pelvic CT images were collected from four medical centers in China. The four DAPSQ trajectory parameters were obtained with the digital measurement method. Finally, the third-generation DAPSQ plate was designed, and its applicability was verified. *Results*: There was no statistically significant difference in the four trajectory parameters when comparing the direct measurement method with the digital measurement method (*p* > 0.05). The average lengths of the pubic region, quadrilateral region, iliac region, and the total length in Chinese population were (60.96 ± 5.39) mm, (69.11 ± 5.28) mm, (84.40 ± 6.41) mm, and (214.46 ± 10.15) mm, respectively. Based on the measurement results, six models of the DAPSQ plate including small size (A1,A2), medium size (B1,B2), and the large size (C1,C2) were designed. The verification experiment showed that all these six type plates could meet the requirement of 94.36% cases. *Conclusions*: A reliable computerized method for measuring irregular pelvic structure was proposed, which not only provided an anatomical basis for the design of the third-generation DAPSQ plate, but also provided a reference for the design of other pelvic fixation devices.

## 1. Introduction

With the rapid development of modern construction and transportation industries, the incidence of acetabular and pelvic fractures caused by high-energy injuries is increasing year by year [1]. Nevertheless, 25% of fractures are also attributed to low-energy trauma, especially in the elderly population [2,3]. Based on the previous literature and our clinical experience, acetabular fractures, particularly complex fractures of acetabulum, often inevitably involved an important anatomical structure; that is, the quadrilateral plate. The quadrilateral plate is located on the medial wall of the acetabulum and plays a critical role of preventing the femoral head from migrating inward into the femur. According to the Letournel—Judet classification, except for simple anterior wall and posterior wall fractures, all other types of acetabular fractures will inevitably involve the quadrilateral plate [4]. Due to its complex anatomy and special location, which is adjacent to major vascular structures and pelvic organs, the surgical treatment of this site is challenging.

Prior to the 1960s, quadrilateral plate fractures were usually treated conservatively with traction, postural reduction, and external orthoses [5,6]. With the development of modern surgical techniques and the in-depth understanding of pelvic and acetabular anatomy, open reduction and internal fixation have become the optimal choice for most quadrilateral plate fractures with significant displacement. The treatment options have evolved from simple pins and screws [7] to cerclage wiring [8] and plates [9,10,11,12,13,14,15]. However, it has to be mentioned that there is a high risk of screws penetrating into the hip joint cavity due to the features of thin and weak bone in the quadrilateral region. To circumvent this problem, many scholars have invented various indirect fixation methods for quadrilateral fracture blocks such as the spring plate [9,10], the quadrilateral surface buttress plate [11,12,13], and the infrapectineal plate [14,15]. Although multiple fixation strategies have been proposed, debates about the best internal fixation methods for this type of fracture continue [16].

Based on our over 20 years of clinical experience, we have designed a novel anterior fixation device for acetabular fractures involving the quadrilateral plate; that is, dynamic anterior plate-screw system for quadrilateral plate (DAPSQ). This fixation device is mainly composed of a special shaped titanium plate including the iliac region plate, the quadrilateral region plate, and the pubic region plate, and several quadrilateral plate screws. Complex acetabular fractures can be treated by a single ilioinguinal approach. Multiple clinical and biomechanical studies conducted by our group have demonstrated the reliability of DAPSQ [17,18,19]. The first-generation DAPSQ is designed based on the conventional 14–16 holes reconstruction plate (Figure 1), which needs temporary shaping during operation. As a result, the final proportion and torsion angle of each plate are totally depended on each operator according to their experiences. In order to form a unified standard and avoid unnecessary time spent for plate shaping during operation, the second-generation standardized titanium plate of DAPSQ with three different models was designed based on cadaveric studies (Figure 2). Our propensity-matched cohort study showed that compared with the first-generation DAPSQ, the second-generation standardized titanium plate has significant advantages of shorter operation time, and lesser intraoperative blood loss and blood transfusion. However, during clinical practice, we have found that the total length of the second-generation plate was not fully suitable for all human anatomy and the proportion of the three regions needed to be further improved.

The most accurate way to learn the anatomical parameters of a certain structure is the direct measurement of cadaveric specimens [20]. Unfortunately, cadaver specimens are very rare in China and the adequate sample size of specimens is extremely difficult to obtain. Of note, the advent of X-Ray, Computed Tomography (CT), and Magnetic Resonance Imaging (MRI) has brought new approaches for anatomical measurements [21,22,23]. After scanning the target structures, indirect anatomical measurements can be made with the aid of digital software. For regular anatomical structures such as long bones of the extremities, accurate measurement is very straightforward to implement. However, as for complicated pelvic anatomy that involves many irregular arc-shaped structures, information on the measurement methods and anatomic parameters is very scarce. In view of this, we considered that it would be necessary to develop a new valuable measure for pelvic anatomy. To our knowledge, this study is the first application of the digital software for large-sample anatomical measurements of DAPSQ plate trajectory. The primary objectives of this study are as follows: (1) to investigate the reliability of the digital measurement method for the DAPSQ plate trajectory; (2) to carry out a digital measurement including the total length and the relative length of each part of DAPSQ trajectory in a large national sample; and (3) to split DAPSQ plate into different models according to the measurement results, so as to match the most anatomical structures of Chinese population.

## 2. Materials and Methods

### 2.1. Direct Measurement of Pelvic Specimens

A total of 22 normal adult pelvic specimens were collected in this study. According to the DAPSQ plate trajectory in the pelvis, the iliac was divided into the pubic region, the quadrilateral region, and the iliac region based on several anatomical landmarks. The specific measurement steps are shown in Figure 3. In brief, the start site of the pubic region was from the pubic symphysis. The selected segmentation point of the pubic region and quadrilateral region was the intersection point (B_PR_, Figure 3a) of the pelvic ring and the extension line of posterior margin in obturator foramen. The segmentation point of the quadrilateral region and iliac region was the projection point (Z_PR_, Figure 3b) of the large sciatic notch on the pelvic ring. The endpoint of iliac region was the iliac crest. Thus, the total length of the DAPSQ plate was the sum of the lengths in the three regions (Figure 3c,d). After determining all the anatomical landmarks, the overall trajectory of the DAPSQ plate was drawn on the pelvis with a marking pen. Then the PDS sutures were placed along the DAPSQ trajectory and an electronic vernier caliper with 0.01 mm sensitivity was used to calculate the total length of the PDS sutures and the respective length of the three regions.

### 2.2. Digital Measurement of the Three-Dimensional (3D) Reconstruction Model of the Pelvis

#### 2.2.1. Pelvic CT Scan and 3D Reconstruction

When direct measurement of the 22 pelvic specimens was completed, all these specimens were scanned using a multidetector CT scanner (TOSHIBA, Aquilion-16, Tokyo, Japan) at the department of Radiology, at the General Hospital of Central Theater Command of the Chinese People’s Liberation Army. Scanning parameters were set as follows: 120 kV, 250 mA, 10 mm layer thickness, and the reconstructed slice thickness was 1.0 mm. The scanned CT data were saved in the digital imaging and communications in medicine (DICOM) format. Then these images were exported from the picture archiving and communication systems (PACS) and imported into the Mimics medical imaging processing software (Version 20.0; Materialise Inc., Leuven, Belgium). The “CT bone segmentation” function of the segment module can be used to obtain the bone tissue of interest quickly. Subsequently, the images were optimized to create a clearer and complete 3D pelvis model.

#### 2.2.2. Digital Morphological Measurement

The detailed procedure of morphological measurements is illustrated in Figure 4. For convenience, the trajectory of the three regions can be seen as three arcs. The length of pubic region was the arc length of OB corresponding to the best-fitting circle obtained by the selection of three points (O, C and B). The lengths of quadrilateral region and iliac region were the arc lengths of BZ and ZH, respectively. Of them, the arc BZ was on a circle formed using points B, Q and Z, while arc ZH was part of the circle created by points Z, E and H. Each point corresponds to a specific anatomic landmark shown in Figure 4. Afterwards, the “distance calculation” function in Mimics software was used to directly obtain the corresponding chord lengths of arcs OB, BZ, and ZH. By using a conversion formula of chord length and arc length as follows: arc length = diameter × arcsin (chord length/diameter), the final lengths of arc OB, BZ, and ZH were calculated. The total length of plate trajectory was the sum of arcs OB, BZ, and ZH. Then, all the 22 pelvic data with digital measurement were compared with data from direct measurement of pelvic specimens.

### 2.3. Collection of Pelvic 3D Reconstruction Data from Different Regions of China

In this study, a total of 504 3D CT image data of the pelvis were collected from four medical centers in northeastern (General Hospital of the Northern Theater Command of the People’s Liberation Army), southern (Foshan Sanshui District People’s Hospital), central (General Hospital of Central Theater Command of the People’s Liberation Army), and northwestern (First Affiliated Hospital of Xinjiang Medical University) China between October 2013 and October 2018. Among them, there were 300 male cases and 204 female cases. The inclusion criteria were one or both side of the pelvis without any lesion, fracture, or anatomical abnormality. Finally, a total of 834 hemipelvises was obtained (Table 1). The present study was approved by the Ethics Committee of our hospital and our use of the cadaveric specimens complied with the institutional and national regulations.

### 2.4. Design and Verification of the Third Generation DAPSQ Plate

#### 2.4.1. Process of Plate Design

Measurements of the 834 hemipelvises were performed as previously described. After 834 measurements, the total length of the plate trajectory was (214.46 ± 10.15) mm. Then we set 214 mm as the total length of the medium model DAPSQ plate. As the length between the two plate holes was approximately 10 mm, we set ±10 mm as the acceptable range for plate use. In other words, the medium model DAPSQ plate will be suitable for patients with the anatomical length of plate trajectory in the range of 204 mm to 224 mm. To achieve the full coverage of the general population, we set 194 mm as the total length of the small model plate and 234 mm as the large model plate. The small model plate will be suitable for patients with anatomical length in the range of 184 mm to 204 mm, while the large model plate could meet needs in the range of 224 mm to 244 mm. In general, these three models of DAPSQ plate (small, medium, and large) could satisfy the requirement ranged between 184 mm to 244 mm.

Beyond the total length of plate trajectory, the proportion of the three regions should be taken into account as well. The length of the pubic region (LPR) was set to 1. The ratio of the length of the iliac region (LIR) to LPR was LIR/LPR. Similarly, the ratio of the length of the quadrilateral region (LQR) to LPR was LQR/LPR. As presented in Figure 5, Pearson’s correlation analysis showed a significant positive correlation between LIR/LPR and LQR/LPR (r = 0.622, *p* < 0.05). This result indicated that LQR/LPR increased with LIR/LPR. Then a linear regression analysis was performed using the LQR/LPR as an independent variable and LIR/LPR as a dependent variable. The linear regression equation was Y = 0.731X + 0.559. The variance analysis of the regression model revealed that the regression equation was statistically significant (*F* = 524.330, *p* < 0.05). According to the distribution characteristics of LQR/LPR, we set LQR/LPR as 1.1 and 1.3 to be the most appropriate values, and the corresponding LIR/LPR was calculated. Therefore, the final ratio of LPR, LQR, and LIR was derived as 1/1.1/1.4 or 1/1.3/1.5. Ultimately, each of the above mentioned three models of DAPSQ plate (small, medium, and large) added two models with different proportion of three regions.

#### 2.4.2. Reliability Verification

To validate plausibility of these newly developed plates, we carried out a verification experiment to match these plates to the 834 pelvises. As already mentioned, we set the error acceptance range of titanium plate in each region as ±10 mm. Taking an A1 titanium plate as an example, the LPR, LQR, and LIR of the type A1 plate were 55 mm, 61 mm, and 78 mm, respectively. Then, we first selected the pelvis with an acceptable range of the pubic region from 45–65 mm, then selected the data where the LQR ranged from 51 mm to 71 mm, and finally matched the data the LIR ranged from 68 mm to 88 mm. By this standard, all the 834 pelvises were matched with the six models one by one, and the total coverage ratio was calculated.

### 2.5. Statistical Analysis

Statistical analysis was conducted using the Statistical Package for the Social Sciences (SPSS) software (Version 22.0, Chicago, IL, USA). Continuous variables were firstly tested for normal distribution and were expressed as mean ± SD with normal distribution and median (and interquartile range) without normal distribution. A paired t test was used to the paired samples. Comparisons between two independent samples were performed by using an independent sample t test. For comparisons of more than two groups, a one-way ANOVA test was used, and then a SNK-q test was used for pairwise comparison. Categorical variables were presented as absolute frequencies (*n*) and percentages (%). Correlation analysis between the two parameters used Person correlation analysis. A *p*-value of less than 0.05 was considered statistically significant.

## 3. Results

### 3.1. Comparison of the Two Measurement Methods

Compared with the direct measurements of pelvic specimens, the lengths of the pubic region, iliac region, and the total were higher than digital measurement, while the length of the iliac region was lower than digital measurement. However, all these differences were not statistically significant. The results are shown in Table 2.

### 3.2. Overall Measurement Results

After 834 hemipelvic measurements, the average lengths of the pubic region, quadrilateral region, iliac region, and the total length in Chinese population were (60.96 ± 5.39) mm, (69.11 ± 5.28) mm, (84.40 ± 6.41) mm, and (214.46 ± 10.15) mm, respectively (Figure 6). In terms of regions, there were 93 hemipelvises collected from northeast China, 549 from central China, 177 from south China, and 15 from northwestern China. When comparing all these trajectory parameters in different regions, cases from northwestern China were excluded due to the small numbers. As shown in Table 3, there were no statistically significant differences when comparing the lengths of the pubic region or iliac region from three Chinese regions (*p* > 0.05). Comparison of the lengths of quadrilateral region or the total length of DAPSQ trajectories among the three groups showed significant difference (*p* < 0.05). Of them, the length of the quadrilateral region in northeast China was significantly longer than that of central China and south China (*p* < 0.05). The length of the quadrilateral region was also longer than that of south China (*p* < 0.05). Similarly, the total length of the DAPSQ trajectories in three Chinese regions showed a similar trend. When considering the gender difference, the analysis was conducted separately for the two genders. Overall, there were significant differences in the four parameters between men and women (*p* < 0.05). Of these, the length of the quadrilateral region, iliac region, and the total length of DAPSQ trajectory in men were significantly longer than women. While the length of pubic region in men was significantly shorter than women (Table 4).

### 3.3. Different Models of Third-Generation DAPSQ Plate and Verification

According to aforementioned methods, a total of six models of the DAPSQ plate (A1, A2, B1, B2, C1, C2) was designed. Detailed parameters of each model are summarized in Table 5. Figure 7 presents the third-generation standardized titanium plates of DAPSQ (a total of 12 plates for the left and right pelvis). The results for the validation experiment showed that the small size plate met the use requirements of 129 (15.47%) cases. Of them, the A1 plate type satisfied the requirement of 112 cases, type A2 plate satisfied the requirement of 104 cases, and 87 cases could use either the A1-type or the A2-type plate. As for the medium size plate, it met the use requirements of 546 (65.47%) cases including 501 cases with type B1, and 424 cases with type B2. There were 378 cases that could use either the B1-type or the B2-type plate. For the large size plate, it met the use requirements of 112 (13.43%) cases. Theoretically, the type C1 plate satisfied 102 cases, type C2 plate satisfied 79 cases, and 69 cases could use either C1-type or C2-type plate. Overall, all the six types of plates were available for a total of 787 cases, accounting for 94.36% of included cases.

## 4. Discussion

### 4.1. The Application of Mimics Software in the Quadrilateral Plate Research

Mimics is a widely used medical image processing and analysis software. By using Mimics software, the reconstructed 3D model on the computer is able to be manually rotated and dynamically observed. The internal anatomical structure of 3D model can be arbitrarily cut, edited, and modified, allowing us to quantitatively analyze the targeting anatomical structure in detail. Multiple previous studies have applied Mimics software to analyze the morphological structure of quadrilateral plate. Zhang et al. [24] have measured the thickness of the quadrilateral plate by using Mimics and proposed the “safe zone” for screw placement and the “danger zone” that is not recommended to place screws. Another digital anatomical measurement conducted by Guo and colleagues has found that the thinner area in the quadrilateral region increased with age, suggesting that the age factor should be considered when designing fixation devices for the quadrilateral area [25]. In addition, Mimics plays an important role in 3D printing technology and virtual surgery simulation [26,27]. Previous studies have found that the application of 3D printing technology could significantly reduce the amount of intra-operative bleeding and operative time [27]. In finite element analysis, Mimics is also an important software to create 3D reconstruction pelvic model for further study of biomechanical stability under different internal fixation methods [28,29,30].

The digital anatomical measurement in this study is precisely based on the generating, editing, and processing functions of Mimics 3D images. Since the DAPSQ plate trajectory is not a regular straight line or curve, in order to facilitate the measurement, we divided the plate trajectory into the pubic region, quadrilateral region, and iliac region, and treated each partition as a circular arc for segmental measurement. Based on the measurement function of the Mimics function list, we could obtain the diameter of the circle where the arc was located and the chord length corresponding to the arc. Finally, the length of the arc could be calculated according to the mathematical formula. Thus, this study brought new lights for measuring the irregular pelvic structures.

### 4.2. DAPSQ Plate Trajectory

DAPSQ is a novel anterior fixation device for acetabular fracture, which is particularly applicable to Chinese people. The biomechanical properties of DAPSQ have been evaluated through series of biomechanical testing experiments including computer simulations using finite element analysis, as well as the application of cadaveric specimens [19,31,32]. DAPSQ takes a unique approach of screw placement, which achieves a direct fixation of quadrilateral plate fracture while completely avoiding the risks of screws mistakenly entering the cavity of hip. In addition, it also creatively provides a less invasive method to solve complex acetabular fracture with a single anterior surgical approach, which supersedes the traditional anterior-posterior combined approach, and has achieved satisfactory clinical outcomes [17]. Through extensive observation of pelvic specimens and postoperative 3D CT images, we have found that the DAPSQ plate is completely adhered on the bone surface, which means the length of DAPSQ plate is equal to the plate trajectory. To facilitate the study, DAPSQ plate is divided into the pubic region, quadrilateral region, and iliac region based on the actual internal function of each part. We defined the plate trajectory starting from the superior border of the pubic symphysis, along the lateral 5 mm of the pelvic ring, up past the anterior border of the sacroiliac joint at the most lateral 10 mm and up along the slightly thickened bone of the medial border of the iliac fossa to the iliac crest. The overall DAPSQ trajectory showed a “S” shape and each part of the plate trajectory was regarded as an arc. This partitioning method is based on the biomechanical properties of these regions. As we stated in our previous studies, before screw insertion, the plates in the pubic region and the iliac region were slightly upturned and did not firmly attach to the bone surface. During the process of the screw insertion, the upturned regions of the DAPSQ plate were firmly attached to the bone surface, which could generate a strong torque force to help quadrilateral screws firmly be attached against the quadrilateral surface [18,19]. Therefore, each part of the DAPSQ plate plays an important role in the successful fixation of this device.

### 4.3. Comparison of Plate Trajectory in Different Chinese Regions

From a morphological point of view, the pelvic morphology varies considerably from geographies to populations. It has been suggested that this difference may be related to topographical and environmental factors. Previous studies showed that from high to low latitudes, the pelvic size showed a descending trend [33]. Moreover, the morphological differences in the pelvis between both sexes are determined mainly by physiological functionality [34]. China has a vast amount of territory with a clear regional gap. This means DAPSQ plate trajectory may be significantly different across regions. Therefore, in the current study, pelvic data were collected from different areas of China. As expected, it was observed that with the increasing of latitudes, the total length of plate trajectory also increased significantly. Compared with the population in south China, the average length of plate trajectory was about 7 mm longer in people from northeast China, suggesting that patients from northeast China are more suitable for the large size model of the DAPSQ plate. It is precisely because of the differences in the DAPSQ trajectory that our measurement results can provide a reference for the design of other quadrilateral plates.

### 4.4. The Design of Third-Generation DAPSQ Plate

In view of the morphological difference in different regions, to achieve a full coverage of the general population in China, we have divided the DAPSQ plate into three different models according to the total length of plate trajectory. Subsequently, each model was further divided into two subtypes according to the proportions of three regions. Through the verification experiment of 834 pelvises, all the six types of plates were available for a total of 787 cases, suggesting that this group of plates could meet the requirement of 94.36% cases. Of note, this verification experiment has only considered from preoperative design perspective. While in the actual application process, the selection criterion of the DAPSQ plate for a certain patient is that the plate length should be shorter or equal to the actual anatomical length. Thus, the actual matching rate of these DAPSQ plates should be much higher than 94.36%. Furthermore, in our previous study, we elaborate on the difference between the first-generation and second-generation DAPSQ plate in the general discussion [18]. In the third-generation DAPSQ plate, we have also made several improvements on the basis of the second-generation one. First of all, the number of plate models has increased from three in the second-generation to six in the third-generation. The current typing is more comprehensive and its specifications are more well-defined. Second, as can be seen from the physical maps of the third-generation DAPSQ plate in Figure 7, the shape of the titanium plate in the quadrilateral region was further improved. There were two rows of screw holes designed in the quadrilateral region. Among them, three additional screw holes were set on the lateral side of the small and medium size plate, and four additional screw holes were set in the large size plate. The use of these additional screw holes in the quadrilateral region has been described in our pervious study. They were mainly used for the insertion of a special instrument so as to help the insertion of quadrilateral screws to the quadrilateral surface. In the second-generation plate, there was only one additional screw hole in this region. The purpose of this design was, on the one hand, to increase the contact region between titanium plate and the bone surface so as to enhance the biomechanical stability of this fixation, and on the other hand, to facilitate the intraoperative operation.

## 5. Conclusions

Digital technology provides a new idea for the large data measurement of the DAPSQ trajectory, and the measurement results could not only provide an anatomical basis for the design of the third-generation DAPSQ plate, but also provide a reference for the design of other pelvic fixation devices. Meanwhile, the diverse design of the third-generation DAPSQ plate could basically meet the clinical needs and further enrich the treatment strategies for complex acetabular fractures involving the quadrilateral area.

## Figures and Tables

**Figure 1 medicina-59-00211-f001:**
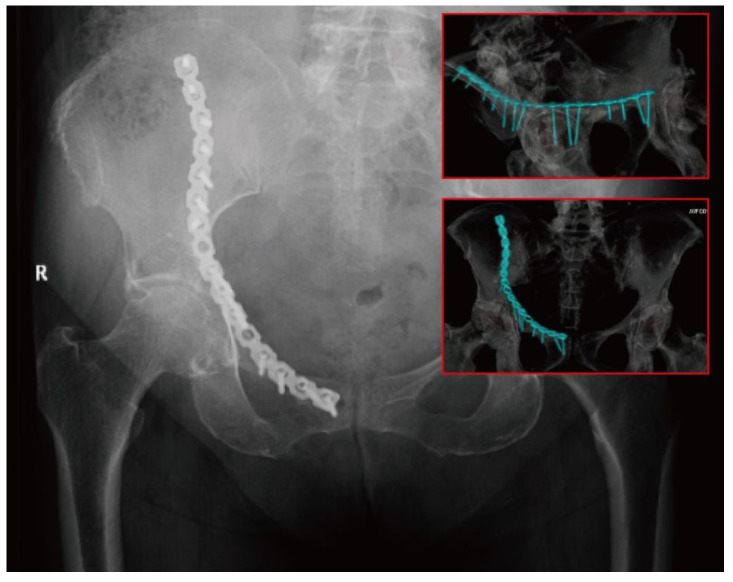
The first-generation dynamic anterior plate-screw system for quadrilateral plate (DAPSQ) for fixation of an acetabular fracture involving quadrilateral plate.

**Figure 2 medicina-59-00211-f002:**
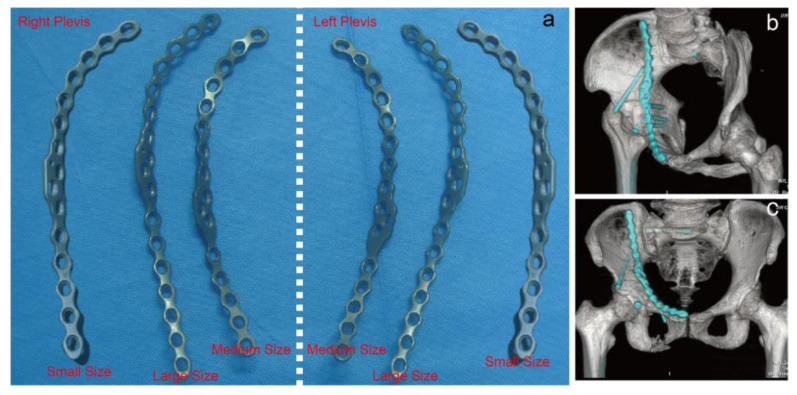
(**a**) Three different models (small size, medium size, large size) of the second-generation DAPSQ plate. (**b**,**c**) The clinical use of the second-generation DAPSQ plate for acetabular fracture.

**Figure 3 medicina-59-00211-f003:**
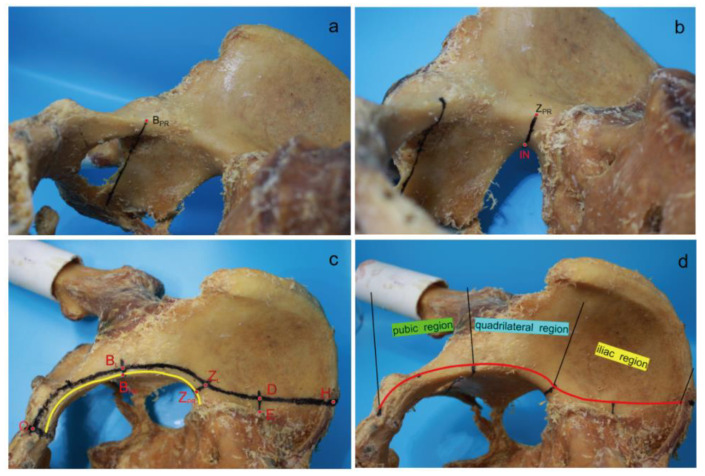
Direct measurement of pelvic specimens. (**a**) A straight line was drawn upward along the posterior margin of obturator foramen to the pelvic ring, and the intersection point was B_PR_. (**b**) A straight line was drawn that was perpendicular to the pelvic ring. This line started from the iliosciatic notch (IN) and met the pelvic ring in the Z_PR_ point. (**c**) The width of 5 mm or 10 mm was determined with a compass. First, several anatomic landmarks on the plate trajectory were identified. Point “O” was the apex of the pubic symphysis. The distances of point “B_L_” to point “B_PR_”, and point “Z_L_” to point “Z_PR”_ were both 5 mm. Point “E” was the lateral edge of the auricular plane, and the distances of point “D” to point “E” was 10 mm. Point “H” was on the iliac crest. After determining all the anatomical landmarks, the overall trajectory of the DAPSQ plate was drawn on the pelvis with a marking pen. Then the PDS sutures were placed along the DAPSQ trajectory and an electronic vernier caliper with 0.01 mm sensitivity was used to calculate the total length of the polydioxanone suture (PDS) sutures and the respective length of the three regions. (**d**) Schematic diagram of DAPSQ plate trajectory and three regions including pubic region (from point “O” to point “BL”), quadrilateral region (from point “B_L_” to point “Z_L_”), iliac region (from point “Z_L_” to point “H”).

**Figure 4 medicina-59-00211-f004:**
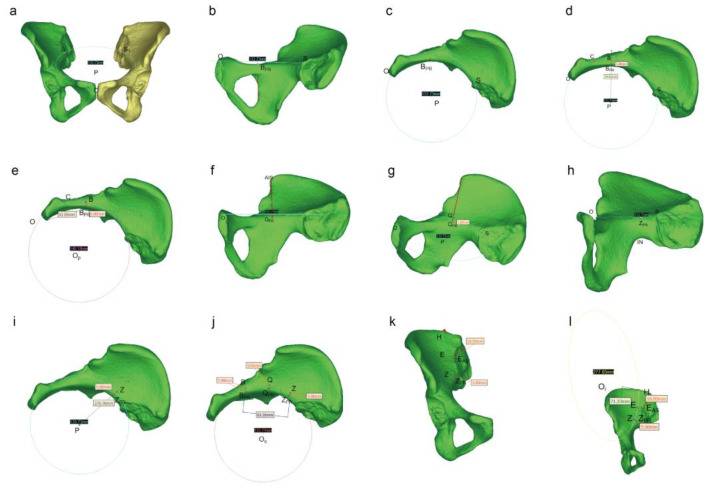
Schematic diagram of the digital measurement process. (**a**) By using the diameter function in the Mimics software, three points including the highest point of the pubic symphysis (point “O”) and the intersection points (point “S” and “S_1_”) of the bilateral arcuate line and the sacroiliac joint (SJ) were selected to obtain the best-fitting circle “P”. The circle P could be regarded as the plane of the pelvic ring. Then hide the mirror image of the hemilateral pelvis after determining the plane. (**b**) Then horizontally move the pelvis and find the projection point (“B_PR_”) of the posterior margin of the obturator foramen on the pelvic ring. (**c**) The top view of the pelvis to determine the point “B_PR_” again. (**d**) Adjust the pelvic position. Make a straight line along the center of the circle P to the point B_PR_, and point “B” was located 5 mm outward the point “B_PR_”. The junction of the pubic tubercle and the pubic comb was point “C”. (**e**) Measurement of the length of the pubic region: Three points including point “O”, “C”, and “B” were used to determine the circle “OP”. The diameter of the circle and straight-line distance from point “O” to point “B” (chord length) can be obtained directly by Mimics software. The length of pubic region was the arc length of OB. By using a conversion formula of chord length and arc length as follows: arc length = diameter × arcsin (chord length/diameter), the final length of arc OB was calculated. (**f**) Rotate this image so that the connecting line between the anterior superior iliac spine (AIS) and the center of circle P is perpendicular to the line OS. Then use the line function of Mimics software to draw a straight line. At this time, the projection point of AIS on the arcuate line was point “Q_PR_”. (**g**) Point “Q” was 5 mm upward along the point QPR. (**h**) Continue to rotate this image until the greater sciatic notch showed the highest point under the plane, and then make a vertical line to the circle P. The intersection point was the projection point (“Z_PR_”) of the greater sciatic notch on the pelvic ring. (**i**) Adjust the pelvic position. Make a straight line along the center of the circle P to the point “Z_PR_”, and point “Z” was located 5 mm outward the point “Z_PR_”. (**j**) Measurement of the length of the quadrilateral region: Three points including point “B”, “Q”, and “Z” were used to determine the circle “Oq”. The diameter of the circle and straight-line distance from point “B” to point “Z” (chord length) can be obtained directly by Mimics software. The length of pubic region was the arc length of BZ. By using a conversion formula of chord length and arc length as follows: arc length = diameter × arcsin (chord length/diameter), the final length of arc BZ was calculated. (**k**) Place the pelvis in an anteroposterior position. Point “E” was located in the lateral side (at a distance 10 mm) of the outermost of auricular surface (point “EAS”). Along the extension of the medial border of the iliac fossa to the highest point of the iliac crest which was regard as point “H”. (**l**) Measurement of the length of the ilium region: Three points including point “Z”, “E”, and “H” were used to determine the circle “Oi”. The diameter of the circle and straight-line distance from point “Z” to point “H” (chord length) can be obtained directly by Mimics software. The length of the pubic region was the arc length of ZH. By using a conversion formula of chord length and arc length as follows: arc length = diameter × arcsin (chord length/diameter), the final length of arc ZH was calculated.

**Figure 5 medicina-59-00211-f005:**
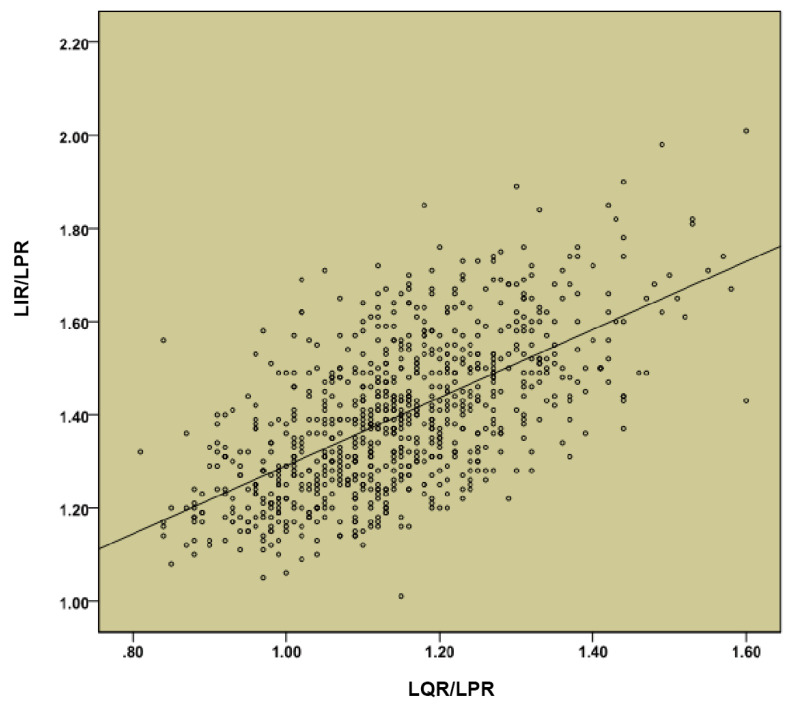
Correlation analysis between length of the iliac region (LIR)/length of the pubic region (LPR) and length of the quadrilateral region (LQR)/LPR.

**Figure 6 medicina-59-00211-f006:**
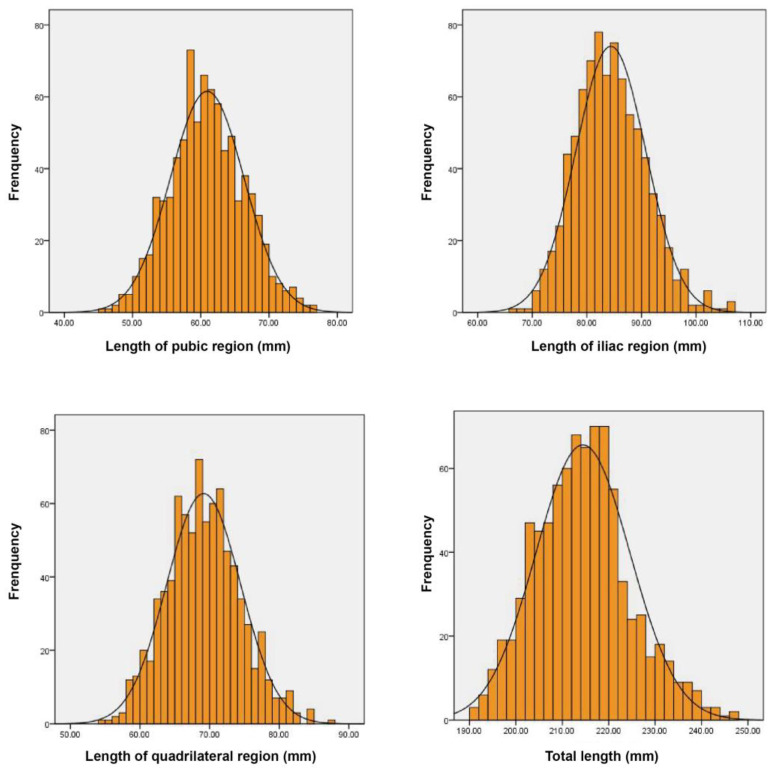
The length distribution of DAPSQ plate trajectory.

**Figure 7 medicina-59-00211-f007:**
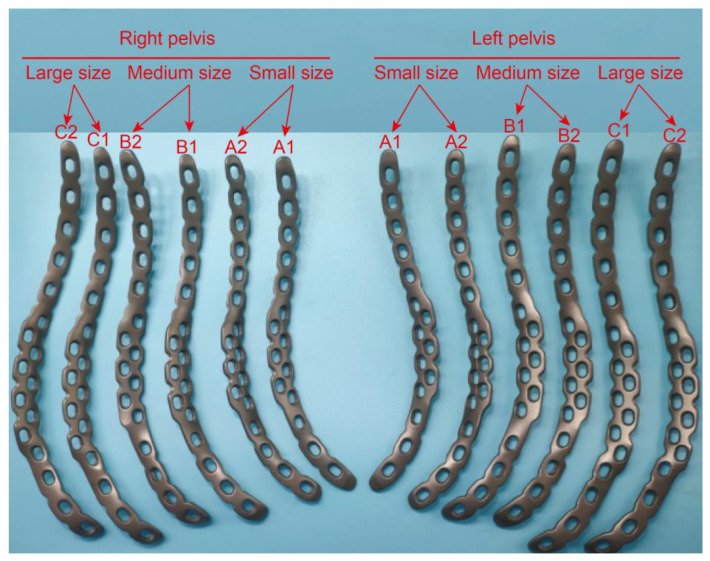
Six different models including small size (A1, A2), medium size (B1, B2) and large size (C1, C2) of the third-generation DAPSQ plate.

**Table 1 medicina-59-00211-t001:** Basic information of pelvic data.

Groups	Age	Intact Pelvis	Left Pelvis	Right Pelvis	Hemilateral Pelvis	Total
Male	41.48 ± 16.18	190	55	55	490	300
Female	47.09 ± 16.53	139	32	34	344	204

**Table 2 medicina-59-00211-t002:** Comparison of direct and digital measurement results (mm).

Groups	Pubic Region	Quadrilateral Region	Ilium Region	Total Length
Direct	60.38 ± 3.90	66.08 ± 3.19	89.19 ± 4.38	215.65 ± 8.23
Digital	60.60 ± 3.79	67.48 ± 4.63	88.20 ± 6.03	216.23 ± 11.41
*t*	0.457	1.988	1.006	0.516
*p*	>0.05	>0.05	>0.05	>0.05

**Table 3 medicina-59-00211-t003:** Anatomical parameters of plate trajectory in different regions of China (mm).

Groups	Pubic Region	Quadrilateral Region	Ilium Region	Total Length
Northeast China (*n* = 93)	61.89 ± 4.55	71.19 ± 5.48	84.94 ± 6.55	218.03 ± 10.14
Central China (*n* = 549)	61.18 ± 5.29	69.12 ± 5.25 *	84.77 ± 6.62	215.07 ± 9.86 *
South China (*n* = 177)	60.03 ± 5.78	67.94 ± 4.98 *#	83.10 ± 5.56	211.07 ± 9.93 *#
*F*-value	4.560	11.868	4.892	17.378
*p*-value	0.11	<0.001	0.08	<0.001

Note: * *p* < 0.05 for comparison with the Northeast China and # *p* < 0.05 for comparison with the Central China.

**Table 4 medicina-59-00211-t004:** Anatomical parameters of plate trajectory in different genders (mm).

Groups	Pubic region	Quadrilateral Region	Ilium Region	Total Length
Male (*n* = 490)	58.64 ± 4.69	69.72 ± 5.17	86.85 ± 6.03	215.20 ± 10.30
Female (*n* = 344)	64.20 ± 4.61	68.31 ± 5.38	80.86 ± 5.19	213.37 ± 9.80
*t*	17.846	5.358	14.506	2.342
*p*-value	<0.001	<0.001	<0.001	0.02

**Table 5 medicina-59-00211-t005:** Parameters of the third-generation DAPSQ plate (mm).

Models		Total Length	Pubic Region	Quadrilateral Region	Iliac Region
Small size	A1	194	55	61	78
	A2	194	51	66	77
Medium size	B1	214	51	67	86
	B2	214	56	73	84
Large size	C1	234	66	74	94
	C2	234	62	80	92

## Data Availability

The dataset used and analyzed during the current study are available from the corresponding author on a reasonable request.

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
