# Peer review of "Third-Generation Dynamic Anterior Plate-Screw System for Quadrilateral Fractures: Digital Design Based on 834 Pelvic Measurements"

_medicina, 2023, doi:10.3390/medicina59020211_

Round 1
Reviewer 1 Report
this study provide a computerized method for measuring irregular pelvic structure was proposed, and provide a reference for the design of other pelvic fixation devices that could be useful in particularly in elderly population.
too much self citation are inserted (5)
I think that at least one or maxium two should be replaced with
- BMC Musculoskelet Disord. 2021 Dec 30;22(Suppl 2):1060. doi: 10.1186/s12891-021-04908-z.
Anterior intrapelvic approach and suprapectineal quadrilateral surface plate for acetabular fractures with anterior involvement: a retrospective study of 34 patients
Gianluca Ciolli, Domenico De Mauro, Giuseppe Rovere, Amarildo Smakaj, Silvia Marino, Lorenzo Are, Omar El Ezzo , Francesco Liuzza
Author Response
Dear Professor,
Thanks very much for taking your time to review this manuscript and giving us an opportunity to revise our manuscript, we appreciate you very much for your positive and constructive comments and suggestions on our manuscript. We have studied your comments carefully and have made revision which marked in red in the paper. We have tried our best to revise our manuscript according to the comments. Point-to-point replies are included as below. I hope that the revision is acceptable for publication in Medicina.
Thank you and best regards.
Yours sincerely
Point 1: This study provide a computerized method for measuring irregular pelvic structure was proposed, and provide a reference for the design of other pelvic fixation devices that could be useful in particularly in elderly population. too much self citation are inserted (5)I think that at least one or maxium two should be replaced with- BMC Musculoskelet Disord. 2021 Dec 30;22(Suppl 2):1060. doi: 10.1186/s12891-021-04908-z. Anterior intrapelvic approach and suprapectineal quadrilateral surface plate for acetabular fractures with anterior involvement: a retrospective study of 34 patients Gianluca Ciolli, Domenico De Mauro, Giuseppe Rovere, Amarildo Smakaj, Silvia Marino, Lorenzo Are, Omar El Ezzo , Francesco Liuzza
Response 1: Thank you for your interest in our fixation technology. We really hope our digital measurement method and fixation device could provide more inspiration for this most changing fracture. We have read the excellent study you suggest for us and it impressed us a lot. Thus we have added it (Ciolli, G.; De Mauro, D.; Rovere, G.; Smakaj, A.; Marino, S.; Are, L.; El Ezzo, O.; Liuzza, F. Anterior Intrapelvic Approach and Suprapectineal Quadrilateral Surface Plate for Acetabular Fractures with Anterior Involvement: A Retrospective Study of 34 Patients. BMC musculoskeletal disorders2021, 22, 1060.) into our reference list [12]. And rearranged the reference order. We are grateful for the suggestion. Thanks for your suggestions again.

Reviewer 2 Report
We thank the authors of this article for the Third-Generation Dynamic Anterior Plate-Screw System for Quadrilateral Fractures. The author introduces the development of the dynamic anterior plate-screw system for quadrilateral plate (DAPSQ). Congratulations on your progress with DAPSQ. The manuscript is generally well-written, however, there are notable deficiencies within the study, some of which were acknowledged by the authors.
Make your abstract more concise. The description for operative indication should be included. The hypothesis of this study is not detailed and unclear in the manuscript. It is the most essential point to introduce this purpose to the authors.
An accurate explanation of the surgical indication is necessary for DAPSQ. It is necessary to explain which surgical approach is used for pelvic operation using the dynamic anterior plate-screw system for quadrilateral plate (DAPSQ), whether the conventional ilioinguinal approach or the modified Stoppa approach. Please tell us how to put and fix these long plates in the pelvis inner cavity.
The 3rd generation DAPSQ seems to have the advantage of being able to perform additional screw fixation on the quadrilateral plate, so it has an advantage over the previous plate. However, it is more important to perform accurate reduction before fixation, while it is important to use an accurately sized plate for pelvic fracture surgery. It is very difficult to reduce the protruding fragment inside the pelvic cavity, and this DASPQ has a design that makes it difficult to screw in from the inside to the outside compared to other companies' plates, so it is assumed that it will not be easy to obtain proper fixation. An explanation of how to perform reduction and fixation using this plate should be added, and additional comments are needed.
Overall well-written paper. Statistical methods are appropriate and sentences are neat.
There is nothing special to modify except the above descriptions. So please revise your manuscript to be convincing. If the editor valued the subject well, I agree with the editor's opinion. Thank you.
Author Response
Dear Editor
Thanks very much for taking your time to review this manuscript and giving us an opportunity to revise our manuscript, we appreciate you and reviewers very much for their positive and constructive comments and suggestions on our manuscript.
We have studied reviewer’s comments carefully and have made revision which marked in red in the paper. We have tried our best to revise our manuscript according to the comments. Attached please find the revised version, which we would like to submit for your kind consideration.
Point-to-point replies are included as below. We would like to re-submit the revised manuscript for your consideration. I hope that the revision is acceptable for publication in your journal.
Thank you and best regards.
Yours sincerely
Point 1: Make your abstract more concise.
Response 1: Thank you for the suggestion. We have streamlined part of the abstract
Abstract: Background and Objectives: To investigate the digital measurement method for the plate trajectory of dynamic anterior plate-screw system for quadrilateral plate (DAPSQ), and then design the third-generation DAPSQ plate that conforms to the needs of Chinese population through performing a large sample anatomical data. Materials and Methods: Firstly, The length of the pubic region, quadrilateral region, iliac region, and the total length of the DAPSQ trajectory were measured by a digital measurement approach in 22 complete pelvic specimens. Then results were compared with the direct measurement of pelvic specimens to verify the reliability of the digital measurement method. Secondly, 504 cases (834 hemilateral pelvis) of adult pelvic CT images were collected from four medical centers of China. The four DAPSQ trajectory parameters were obtained with digital measurement method. Finally, the third-generation DAPSQ plate was designed and its applicability was verified. Results: There was no statistically significant difference in the four trajectory parameters when comparing the direct measurement method with the digital measurement method (P>0.05). The average lengths of the pubic region, quadrilateral region, iliac region, and the total length in Chinese population were (60.96±5.39) mm, (69.11±5.28) mm, (84.40±6.41) mm and (214.46±10.15) mm, respectively. Based on the measurement results , six models of DAPSQ plate including small size (A1, A2), medium size (B1, B2) and large size (C1, C2) was designed. The verification experiment showed that all these six type plates could meet the requirement of 94.36% cases. Conclusions: A reliabl computerized method for measuring irregular pelvic structure was proposed, which not only provide an anatomical basis for the design of ethe third-generation DAPSQ plate, but also provide a reference for the design of other pelvic fixation devices.(please see Line 25-46).
Point 2:The description for operative indication should be included. An accurate explanation of the surgical indication is necessary for DAPSQ.
Response 2:Thank you for the suggestion. We have described DAPSQ operative indications in published article.
“DAPSQ is more suitable for displaced acetabular fractures characterized by anterior column injuries, and fractures involving the quadrilateral plate, such as both columns mainly with anterior column injury, transverse fractures with anterior displacement, partial anterior column posterior hemi‐transverse, and T‐shaped fractures. For old acetabular fractures, acetabular posterior wall fractures, or in cases of both columns mainly with posterior column injury, an anterior–posterior surgical approach is often needed. Moreover, due to the limited compression and deformation of the reconstruction plate after screw placement, DAPSQ cannot be applied to patients with severe osteoporosis”
Wu H, Shang R, Cai X, Liu X, Song C, Chen Y. Single Ilioinguinal Approach to Treat Complex Acetabular Fractures with Quadrilateral Plate Involvement: Outcomes Using a Novel Dynamic Anterior Plate-Screw System. Orthop Surg. 2020;12(2):488-497. doi:10.1111/os.12648
Point 3:The hypothesis of this study is not detailed and unclear in the manuscript. It is the most essential point to introduce this purpose to the authors.
Response 3: Thank you for the suggestion. We have introduced purposes as follows :“The primary objectives of this study are as follows: (1) to investigate the reliability of the digital measurement method for DAPSQ plate trajectory; (2) to carry out a digital measurement including the total length and the relative length of each part of DAPSQ trajectory in a large national sample; (3) to split DAPSQ plate into different models according to the measurement results, so as to match the most anatomical structures of Chinese population.”(please see the red font in Line 131-136).
Point 4: It is necessary to explain which surgical approach is used for pelvic operation using the dynamic anterior plate-screw system for quadrilateral plate (DAPSQ), whether the conventional ilioinguinal approach or the modified Stoppa approach.
Response 4:Thank you for the suggestion. We usually perform the surgery with the ilioinguinal approach.(please see the red font in Line 84)
Point 5: Please tell us how to put and fix these long plates in the pelvis inner cavity.
Response 5:Thank you for the suggestion. The ilioinguinal approach have three”anatomical windows”. After opening the three anatomical windows of the iliogroin approach, the plate is inserted along the suprapubic branch from the third window, and through the second to the first. Place the internal fixation along the pelvic ring and adjust the internal fixation posture. Response 6 explained the fixation process.
Point 6: The 3rd generation DAPSQ seems to have the advantage of being able to perform additional screw fixation on the quadrilateral plate, so it has an advantage over the previous plate. However, it is more important to perform accurate reduction before fixation, while it is important to use an accurately sized plate for pelvic fracture surgery. It is very difficult to reduce the protruding fragment inside the pelvic cavity, and this DASPQ has a design that makes it difficult to screw in from the inside to the outside compared to other companies' plates, so it is assumed that it will not be easy to obtain proper fixation. An explanation of how to perform reduction and fixation using this plate should be added, and additional comments are needed.
Response 6: Thank you for the suggestion. Since the purpose of this article is to design the third generation DAPSQ, we do not discuss how to place and fix the fixation in this article. We have published the installation details of the first generation DAPSQ and the second generation DAPSQ.
“The key surgical steps were as follows: two or more fixation screws on the iliac and pubic region should first be fixed to stabilize the acetabular anterior column. Then, with the help of a 4.5-mm screwdriver, quadrilateral screws were inserted along the pelvic brim and parallel to the surface of the quadrilateral plate under direct vision, and only the 1/3 to 1/2 transverse diameter of the quadrilateral screw was screwed into the bone to avoid penetrating the hip. And during the process of screw insertion, the torsion and elastic recoil of the plate could provide a strong holding force for quadrilateral screws to block the inward displacement of the quadrilateral plate. Also, make sure the distal end of the quadrilateral screws extended at least 10 mm beyond the fracture line.”
Wu H, Shang R, Liu X, Song C, Chen Y, Cai X. A novel anatomically pre-contoured side-specific titanium plate versus the reconstruction plate for quadrilateral plate fractures of the acetabulum: a propensity-matched cohort study. J Orthop Surg Res. 2020;15(1):172. Published 2020 May 14. doi:10.1186/s13018-020-01659-w
